

# Risk and clinical predictors of osteoporotic fracture in East Asian patients with chronic obstructive pulmonary disease: a population-based cohort study

Ping-Hsueh Lee[1,2], Victor C. Kok[3,4,5], Po-Liang Chou[6], Ming-Chang Ku[2,7], Yu-Ching Chen[4,5] and Jorng-Tzong Horng[4,5,6]

[1] Department of Geriatric Medicine, Kuang Tien General Hospital, Taichung, Taiwan

[2] Jen-Te Junior College of Medicine, Nursing and Management, Miaoli County, Taiwan

[3] KTGH Cancer Center, Kuang Tien General Hospital, Taichung, Taiwan

[4] Department of Bioinformatics and Medical Engineering, Asia University Taiwan, Taichung, Taiwan

[5] Disease Informatics Research Group, Asia University Taiwan, Taiwan

[6] Department of Computer Science and Information Engineering, National Central University, Jhongli, Taoyuan, Taiwan

[7] Department of Diagnostic Radiology, Kuang Tien General Hospital, Taichung, Taiwan

Corresponding authors
Victor C. Kok, victorkok@asia.edu.tw
Jorng-Tzong Horng,
horng@db.csie.ncu.edu.tw

## ABSTRACT

**Introduction**. Osteoporosis is becoming an impending epidemic in the Asia-Pacific region. The association between risk of osteoporotic fracture (OTPF) and chronic obstructive pulmonary disease (COPD) in East Asian patients is yet to be fully examined. We conducted a nationwide population-based retrospective cohort study of 98,700 patients aged $\geq$50 years with or without COPD using a national administrative claims dataset.

**Materials and Methods**. The patients were divided into COPD and comparison groups comprising 19,740 and 78,960 patients, respectively. The groups were 1 to 4 matched for age, gender, index date, diabetes mellitus, pre-existing osteoporosis and chronic kidney disease. Information such as the geographic area where southern part represented more sunshine exposure, smoking-related diagnoses, alcohol use disorder, whether there was regular use of inhaled corticosteroids and oral corticosteroids, vitamin D prescriptions, Charlson-Deyo comorbidity index score, and other relevant medical comorbidities were extracted for analysis. They were followed up until OTPF or the end of the year 2013. The outcome measure was an osteoporotic vertebral fracture and other long-bone fractures. A multivariate Cox model was constructed to derive adjusted hazard ratios (aHR) for OTPF with corresponding 95% confidence intervals (CI) after controlling for age, sex, insurance premium category, vitamin D prescription, osteoporosis, and coronary heart disease (CHD). Kaplan–Meier curves of the probability of OTPF-free survival for each cohort were compared using the log-rank test. Patients with OTPF during the first follow-up year were excluded from the overall risk calculation. Contributing factors to the increased risk of OTPF in COPD patients were examined in a sensitivity analysis.

**Results**. After a total follow-up of 68,743 patient-years for the COPD group and 278,051 patient-years for the matched comparison group, the HR for OTPF was 1.24
(95% CI [1.02–1.51]; $P = 0.0322$) in COPD patients. The aHR was increased by 30% for vertebral OTPF (aHR = 1.297, 95% CI [1.020–1.649]; $P = 0.0339$). Differential lag time sensitivity analysis revealed a progressively elevated risk up to 8-fold increase in women (aHR = 8.0 (95% CI [1.81–35.4]; $P < 0.01$)) during the fifth follow-up year. COPD patients with pre-existing osteoporosis or given vitamin D prescription harbor a sustained increased risk up to the 5th (aHR, 4.1; 95% CI [1.61–10.35]) and third (aHR, 2.97; 95% CI [1.48–5.97]) follow-up year, respectively.

**Conclusions**. Our nationwide population-based cohort study demonstrates that East Asian COPD patients aged 50 and beyond do harbor a modestly increased risk for osteoporotic vertebral fractures particularly for those who are female, have pre-existing osteoporosis or require vitamin D prescription.

## INTRODUCTION

Osteoporosis is the most common bone disease in the world and is characterized by low bone mass and derangement of bone microarchitecture. The prevalence of osteo porotic fracture (OTPF) was estimated to be 9.0 million worldwide in the year 2000 (*Johnell & Kanis, 2006*). In a cohort of Chinese women, the prevalence rates of primary osteoporosis affecting the spine, femoral neck, and hip were reported to be 32%–34%, 16.3%, and 18.9%, respectively (*Wu et al., 2004*). In other Asian countries, the prevalence rate of osteoporosis in women ≥50 years has been reported as 34% in Korea and 31% in Japan (*Choi et al., 2012*; *Iki et al., 2001*). OTPF results in significant reductions in quality of life due to pain, depression, and disability (*Poole & Compston, 2006*). Elderly patients who sustain nondisplaced hip fractures experience predictable and lasting loss of function, particularly in patients with pulmonary disease. Both age and chronic obstructive pulmonary disease (COPD) affect the speed of functional recovery (*Eisler et al., 2002*). Furthermore, the mortality rate following hip fracture reportedly was as high as 36% in a systemic epidemiological review (*Abrahamsen et al., 2009*). The estimated cost of treatment of osteoporosis-related fracture in the United States was approximately $17 billion in 2005 and is estimated to increase to $25.3 billion by 2025 (*Burge et al., 2007*).

Despite the substantial clinical burden of osteoporosis, treatment rates are low in some East Asian countries. In a national screening program in Korea, the estimated treatment rate was only 14.4% in osteoporotic women (*Choi et al., 2012*). Because low bone mineral density (BMD) is a major risk factor for fractures, timely treatment can prevent this disastrous outcome (*Nelson et al., 2010*). Therefore, early identification of high-risk patients is an important step toward increasing treatment rates. However, relying on BMD screening alone is not considered sufficient to detect high-risk patients in a timely and cost-effective manner (*Ogura-Tomomatsu et al., 2012*; *Schuit et al., 2004*).

Patients with COPD have many shared features with osteoporotic patients, such as low body mass index (BMI), inactivity, and chronic steroid use. COPD has been shown to be

associated with low BMD in recent studies (*Jaramillo et al., 2015*; *Looker, 2014*). However, the association between risk of OTPF and COPD has not been well studied, particularly in the Asian population. The prevalence of osteoporosis has been shown to be higher in COPD patients compared with controls (*Graat-Verboom et al., 2009*; *Schnell et al., 2012*), and the prevalence of hip fracture was reportedly higher in a cross-sectional study of 465 COPD patients in Brazil (*Kulak et al., 2010*). To date, no studies have been reported from East Asian countries regarding the magnitude of the risk of subsequent OTPF contributed by each clinical predictor in COPD patients. The aim of the present study was to evaluate the association between COPD and risk of OTPF and identify clinical predictors of fracture in East Asian patients. The magnitude of the risk of OTPF contributed by clinical predictors would be calculated.

## MATERIALS AND METHODS

### Data source

Data from 2007 to 2013 was obtained from the Longitudinal Health Insurance Dataset (LHID) of the National Health Insurance Research Database (NHIRD). The National Health Insurance (NHI) program, established in 1995 in Taiwan, is a mandatory health insurance program requiring every resident, from a newborn to an elderly person, to join. It provides comprehensive insurance coverage for medical services through outpatients, inpatients, and emergency departments, in addition to traditional Chinese medicine. The program currently has a coverage rate of 99%. LHID consists of 1 million beneficiaries randomly sampled from NHIRD. The dataset includes NHI enrolment files and claims data, such as examinations, medical procedures, drug prescriptions, and diagnoses, of all included patients. The International Classification of Diseases, Ninth Revision, Clinical Modification codes (ICD-9-CM codes; 2001 revision) were used by physicians to code diseases in the system. All recognizable personal information was encrypted according to regulations specified by the Bureau of NHI. The authors have successfully conducted several population-based retrospective cohort studies using the LHID (*Kok et al., 2014*; *Kok et al., 2016*; *Kok et al., 2015a*; *Kok et al., 2015b*). The present study was approved by the accredited local in-house Institutional Review Board with a certificate number KTGH-IRB 10520. The IRB approved that no any form of consent (Verbal/Written) was required for this study.

### Study design

We conducted a population-based, observational, cohort study to assess the association between COPD and risk of OTPF. All patients diagnosed with COPD between January 2007 and December 2013 were identified (ICD-9 CM: 490–492, 496). The index date was defined as the month in which the diagnosis of COPD was made. Exclusion criteria were as follows: (1) age <50 years; (2) less than two COPD outpatient claims within 1 year with no inpatient records; and (3) pre-existing COPD or OTPF before January 1, 2007. The control group was composed of the remaining patients in LHID without COPD, with 1 to 4 matched for age, gender, and index month, diabetes mellitus, pre-existing osteoporosis and chronic kidney disease. The exclusion criteria applied to the COPD

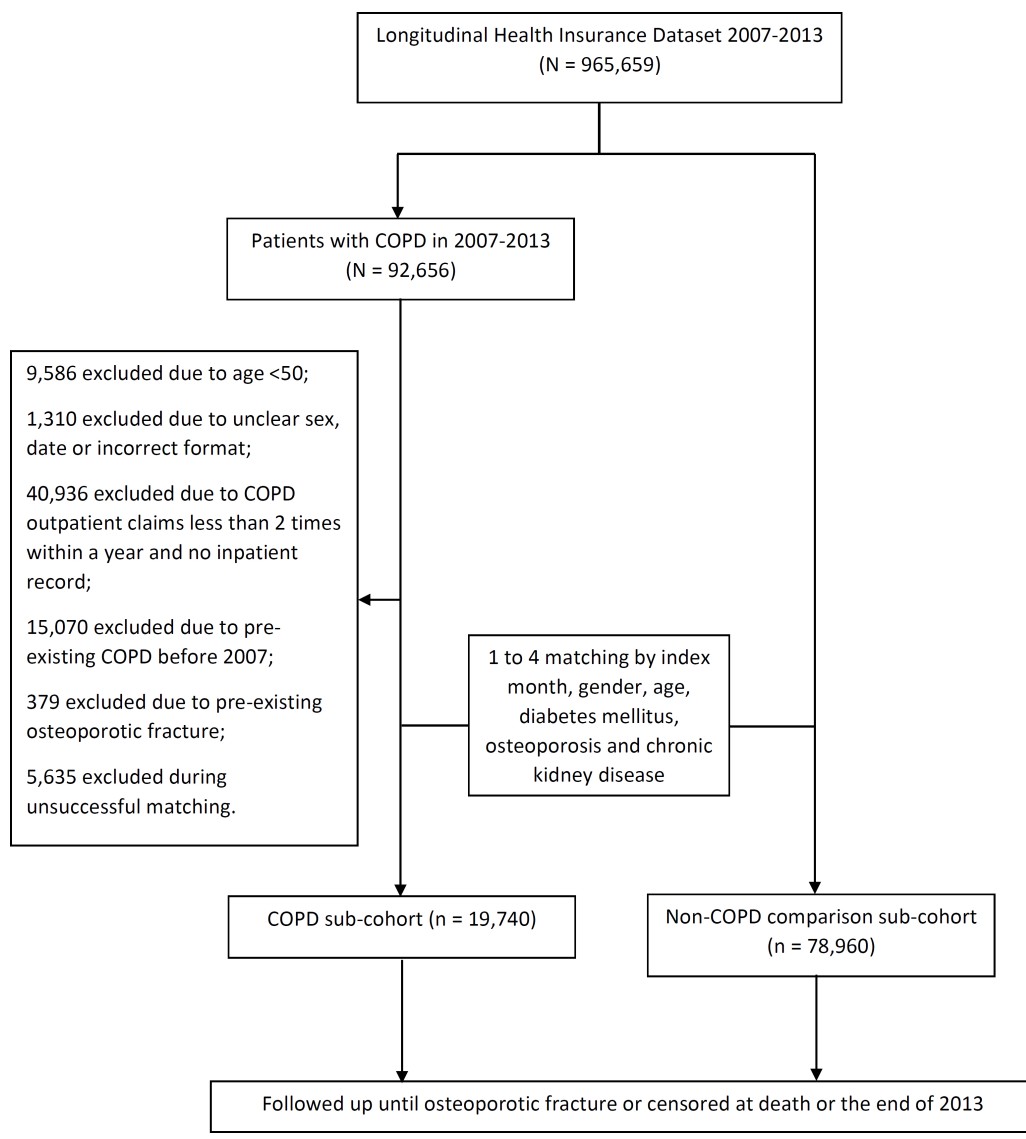

**Figure 1  Consort diagram of the present study flow.**

group was used for the control group. All study patients were followed up until the first incidence of OTPF which was further separated into vertebral fractures and the other long bone fractures, drop-out from the insurance program, or the end of the present study (Fig. 1).

## Data extraction

Patients age, gender, date of accrual, geographic residential area, insurance premium category, behavioral proxies such as smoking related diagnoses and alcohol use disorders; medical comorbidities including diabetes mellitus, pre-existing osteoporosis, chronic kidney disease, rheumatoid arthritis, hypertension, dyslipidemia, coronary heart disease, chronic liver disease, stroke and dementia; medical prescriptions such as vitamin D,

inhaled corticosteroids (ICS: budesonide, beclomethasone, ciclesonide, fluticasone) and oral corticosteroids (OCS: cortisone acetate, dexamethasone, fludrocortisone, methylprednisolone, prednisolone, triamcinolone); and Charlson-Deyo Comorbidity Index Score were collected from the dataset and calculated. Longitudinal tracking data of study participants until the occurrence of a specific type of osteoporotic fracture, e.g., vertebral fracture or femoral neck fracture were extracted. Candidates who would be included were also screened for pathological fractures due to cancer metastases (ICD-9-CM code 198.5), renal osteodystrophy (588.0) and secondary hyperthyroidism (588.81).

## Outcome measures

The primary outcome measure of the present study was OTPF, defined as any pathological fracture due to osteoporosis (ICD-9-CM codes: 733.0x + 733.1x) documented in inpatient or outpatient data files during the study period. Osteoporotic vertebral fracture has a unique code as 733.13. To prevent overstated or false claiming, NHI randomly and regularly reviews claims data, including patient history and laboratory and imaging reports. Only cases with correct coding for all diagnoses are eligible for full reimbursement. Any violation of the coding regulations results in payment retrieval plus a punitive fine, which may be several times the original payment. To ensure accuracy, all coding procedures were performed by groups of specialists assigned by contributing hospitals. OTPF in the present study was defined as any new and matched ICD-9 coding of cohort subjects between 2007 and 2013 in outpatient, emergent, or inpatient settings. Therefore, we believe that all events that occurred during the follow-up period were recorded with a high degree of accuracy. Furthermore, we excluded pathological fractures that resulted from cancer metastasis, renal osteodystrophy, or secondary hyperparathyroidism (ICD-9-CM codes: 733.1 + (198.5, 588.0, or 588.1)) (Table S1). Patients with OTPF during the first follow-up year were excluded from the overall risk calculation.

## Covariates and confounders

Socioeconomic status of participating subjects was approximated using different category by insurance premium. Exposure to different levels of sunshine was also taken into account by the use of the residential area of the participating subjects (more sunshine in the southern part of the country). Lifestyle factors included smoking-related diagnoses and alcohol used disorder were used as proxies for cigarette smoking and alcohol drinking were extracted. Medication history of inhaled corticosteroid and/or oral corticosteroid use and vitamin D prescription were extracted from the dataset. Comorbidities were identified according to ICD-9-CM codes, including rheumatoid arthritis, hypertension, dyslipidemia, coronary heart disease (CHD), liver disease, stroke, and dementia. An overall score of the Charlson-Deyo comorbidity index of each patient was collected.

## Statistical analysis

Patient-year data and incidence rates were evaluated. A multivariate Cox model was constructed to derive adjusted hazard ratios (aHR) with corresponding 95% confidence intervals (CI) after controlling for factors that were revealed significant in the univariate Cox model so that overadjustment (overfitting) and noise could be avoided. Fracture-free

survival was assessed using the Kaplan–Meier analysis. Survival curves were compared between COPD and non-COPD groups using the log-rank test. Differential lag time sensitivity analysis was carried out to examine the effect of different exposure time lag on the risk of osteoporotic fractures among COPD patients who possessed the characteristic such as female and specific medical comorbidity. We used the bivariate Cox model to evaluate the risk of OTPF in COPD when coupled with each medical comorbidity. $P$-values of $<0.05$ were considered statistically significant. All study analyses were performed using the SPSS statistical software (IBM SPSS Statistics Version 22).

## RESULTS

### Characteristics of the study population

From 2007 to 2013, 92,656 newly-diagnosed COPD patients were identified from LHID ($n = 965,659$). Among these, 70,191 were excluded after applying the exclusion criteria described above. Reasons for exclusion were as follows: incomplete data registration ($n = 1,310$) and unsuccessful matching ($n = 5,635$). As a result, 19,740 patients were recruited to the COPD sub-cohort and 78,960 to the non-COPD sub-cohort after 1 to 4 matching for age, gender, index month, diabetes mellitus, pre-existing osteoporosis and chronic kidney disease. The mean age (and standard deviation) in the COPD group was 66 years (10.23), with 62.5% of patients $<70$ years of age. COPD patients were predominantly male (55.8%). The mean follow-up duration was 2.97 years (standard deviation (SD), 2.06 years) in the COPD group and 3.01 years (SD, 2.05 years) in the non-COPD group, which were not statistically significantly different ($P = 0.0949$). Hypertension and hyperlipidemia were the most commonly observed co-morbidities. Except for dyslipidemia, all non-matching co-morbidities, such as rheumatoid arthritis, hypertension, and coronary heart disease, were differently distributed between the two groups. The mean Charlson Comorbidity Index (CCI) score was significantly higher in the COPD sub-cohort (1.14 vs. 1.02; $P < 0.0001$; Table 1). More patients with COPD were classified as a regular user of oral prednisolone (16.9% vs. 6.4%; $P < 0.001$; Table 1).

### Incidence of OTPF

There were 131 OTPF events in the COPD group during the study period (68,743 patient-years), with an incidence rate of 1.91/1,000 patient-years. In the comparator group matched for age, gender, index month, diabetes mellitus, chronic kidney disease and pre-existing osteoporosis, there were 429 OTPF events in 278,051 patient-years, with an incidence rate of 1.54/1,000 patient-years. Patients with COPD were significantly more likely to have OTPF (crude hazard ratio (HR), 1.24; 95% CI [1.02–1.51]; $P = 0.0322$; Table 2), as demonstrated by separation of the two cumulative incidence curves (Fig. 2) and osteoporotic fracture-free survival curves in the Kaplan–Meier analysis (Fig. S1).

After multivariate Cox proportional hazards regression controlling for sex, age, vitamin D prescription, insurance premium category, pre-existing osteoporosis and coronary heart disease, the difference between the two groups lost statistical significance (aHR, 1.21; 95% CI [0.992–1.469]; $P = 0.0597$). However, when the outcome was a vertebral

**Table 1** Demographic characteristics of COPD patients and Non-COPD patients 1-to-4-matched by age, sex, index date, diabetes mellitus, pre-existing osteoporosis, and chronic kidney disease.

| Descriptor | COPD patients | | Non-COPD patients | | P value |
|---|---|---|---|---|---|
| | N = 19,740 | (%) | N = 78,960 | (%) | |
| **Age mean (SD)** | 66.02 (10.23) | | 65.92 (10.23) | | 1 |
| **Age group** | | | | | |
| 50 ∼ 59 | 6,423 | 32.54 | 26,140 | 33.11 | |
| 60 ∼ 69 | 5,915 | 29.96 | 23,501 | 29.76 | |
| 70 ∼ 79 | 5,077 | 25.72 | 20,215 | 25.60 | |
| >80 | 2,325 | 11.78 | 9,104 | 11.53 | |
| **Gender** | | | | | 1 |
| Female | 8,732 | 44.24 | 34,928 | 44.24 | |
| Male | 11,008 | 55.76 | 44,032 | 55.76 | |
| **Other matched characteristics** | | | | | |
| Diabetes mellitus | 4,726 | 23.94 | 18,904 | 23.94 | 1 |
| Osteoporosis | 1,187 | 6.01 | 4,748 | 6.01 | 1 |
| Chronic kidney disease | 2,746 | 13.91 | 10,984 | 13.91 | 1 |
| **Follow-up (Year)/Mean (SD)** | 2.97 (2.06) | | 3.01 (2.05) | | 0.949 |
| 1 | 3,051 | 15.46 | 11,618 | 14.71 | |
| 2 | 2,976 | 15.08 | 11,707 | 14.83 | |
| 3 | 2,820 | 14.29 | 11,355 | 14.38 | |
| 4 | 2,575 | 13.04 | 10,442 | 13.22 | |
| 5 | 2,523 | 12.78 | 10,309 | 13.06 | |
| 6 | 2,597 | 13.16 | 10,754 | 13.62 | |
| 7 | 3,170 | 16.06 | 12,691 | 16.07 | |
| **Residential Area** | | | | | <.0001 |
| North | 8,386 | 42.48 | 34,410 | 43.58 | |
| Central | 5,300 | 26.85 | 19,132 | 24.23 | |
| South | 5,330 | 27.00 | 22,655 | 28.69 | |
| East | 591 | 2.99 | 1,934 | 2.45 | |
| **Smoking-related diagnoses** | 12,353 | 62.58 | 2,772 | 3.51 | <0.0001 |
| **Alcohol use disorder** | 598 | 3.03 | 1,557 | 1.97 | <0.0001 |
| **Insurance premium category** | | | | | <0.0001 |
| <15,000 NTD | 10,041 | 50.87 | 39,186 | 49.63 | |
| 15,000—21,999 NTD | 6,028 | 30.54 | 23,634 | 29.93 | |
| ≥22,000 NTD | 3,671 | 18.60 | 16,140 | 20.44 | |
| **Comorbidities** | | | | | |
| Rheumatoid Arthritis | 545 | 2.76 | 1,790 | 2.27 | <0.0001 |
| Hypertension | 11,509 | 58.30 | 44,266 | 56.06 | <0.0001 |
| Dyslipidemia | 6,813 | 34.51 | 27,511 | 34.84 | 0.3868 |
| Coronary heart disease | 5,528 | 28.00 | 17,192 | 21.77 | <0.0001 |
| Liver disease | 3,824 | 19.37 | 13,644 | 19.28 | <0.0001 |
| Stroke | 3,994 | 20.23 | 12,188 | 15.44 | <0.0001 |
| Dementia | 1,144 | 5.80 | 2,481 | 3.14 | <0.0001 |

| Descriptor | COPD patients | | Non-COPD patients | | P value |
|---|---|---|---|---|---|
| | N = 19,740 | (%) | N = 78,960 | (%) | |
| **Type of ICS** | | | | | |
| Budesonide | 1,214 | 6.17 | 440 | 0.56 | <0.0001 |
| Beclomethasone | 267 | 1.36 | 169 | 0.21 | <0.0001 |
| Ciclesonide | 46 | 0.23 | 16 | 0.02 | <0.0001 |
| Fluticasone | 1,762 | 8.96 | 425 | 0.54 | <0.0001 |
| **Type of OCS** | | | | | |
| Cortisone acetate | 62 | 0.32 | 125 | 0.16 | <0.0001 |
| Dexamethasone | 325 | 1.65 | 770 | 0.98 | <0.0001 |
| Fludrocortisone | 18 | 0.09 | 51 | 0.06 | <0.0001 |
| Methylprednisolone | 373 | 1.90 | 656 | 0.83 | <0.0001 |
| Prednisolone | 3,330 | 16.93 | 5,012 | 6.36 | <0.0001 |
| Triamcinolone | 30 | 0.15 | 97 | 0.12 | <0.0001 |
| **Vitamin D prescription** | 487 | 2.48 | 1,689 | 2.14 | 0.0045 |
| **CCI score[a]/Mean (SD)** | 1.14 (1.36) | | 1.02 (1.32) | | <0.0001 |
| CCI score 0 | 8,604 | 43.59 | 38,699 | 49.01 | |
| CCI score 1, 2 | 8,051 | 40.79 | 29,345 | 37.16 | |
| CCI score 3, 4 | 2,308 | 11.69 | 8,235 | 10.43 | |
| CCI score ≥ 5 | 777 | 3.94 | 2,681 | 3.40 | |

**Notes.**

[a] COPD was excluded from the CCI score in the COPD cohort.

CCI, Charlson-Deyo comorbidity index; ICS, inhaled corticosteroid; NTD, New Taiwan Dollar; OCS, oral corticosteroid; SD, standard deviation.

OTPF, both crude HR (1.33; 95% CI [1.05–1.69]; $P = 0.0189$) and aHR (1.297; 95% CI [1.020–1.649]; $P = 0.00339$) were statistically significant. Fig. 3 demonstrates the clear separation of the two cumulative incidence curves. Owing to few events as long bone fracture, the crude HRs were non-significantly increased, for example, in ulnar fractures and femoral neck fracture (Table 2).

It is noteworthy that COPD patients who were older (up to 7.5-fold increase of the risk in terms of aHR in patients aged >80 years as compared with the age group of 50–59 years), female (3.8-fold increase), who received vitamin D prescriptions (3.3-fold increase), with pre-existing osteoporosis (3-fold increase) or with coronary heart disease (1.5-fold increase) had higher risk for OTPF than their non-COPD comparators (Table 2). In sensitivity analysis based on differential lag time with vertebral OTPF as the outcome, female COPD patients had a sustained and progressively increased risk of vertebral OTPF compared with non-COPD females from the second (aHR = 3.3; 95% CI [1.89–5.74]; $P < 0.0001$) to fifth (aHR = 8.0; 95% CI [1.81–35.4]; $P < 0.01$) follow-up year. In COPD patients with pre-existing osteoporosis, the risk of vertebral OTPF, approximately 3- to 4-fold increased, sustained through to the fifth follow-up year. COPD patients who were co-prescribed vitamin D had an statistically significantly increased risk in the second (aHR = 3.62; 95% CI [2.1–6.23]; $P < 0.0001$) and third (aHR = 2.97; 95% CI [1.48–5.97]; $P < 0.01$) follow-up year (Table 3). Comorbidities, such as rheumatoid arthritis, diabetes mellitus, hypertension, dyslipidemia, CKD, liver disease, stroke, or

**Table 2** Incidence of osteoporotic fractures for COPD patients compared with non-COPD patients and crude and adjusted hazard ratio for an episode of osteoporotic fracture.

| Variables | COPD patients | | | Non-COPD patients | | | Incidence Rate Ratio (95 % CI) | Crude HR | Adjusted HR[a] |
|---|---|---|---|---|---|---|---|---|---|
| | Event | PY | Rate | Event | PY | Rate | | | |
| All | 131 | 68,743.3 | 1.91 | 429 | 278050.7 | 1.54 | **1.24 (1.01–1.51)** | **1.24 (1.02–1.51)** (P = 0.0322) | 1.207 (0.992–1.469) (P = 0.0597) |
| Vertebral fractures | 89 | 68743.3 | 1.29 | 271 | 278050.7 | 0.97 | **1.33 (1.03–1.69)** | **1.33 (1.05–1.69)** (P = 0.0189) | **1.297 (1.020–1.649)** (P = 0.0339) |
| Ulna fractures | 1 | 68743.3 | 0.01 | 3 | 278050.7 | 0.01 | 1.35 (0.03–16.79) | 1.35 (0.14–12.99) (P = 0.7943) | |
| Neck of femur fractures | 4 | 68743.3 | 0.06 | 7 | 278050.7 | 0.03 | 2.31 (0.50-9.09) | 2.319 (0.68–7.92) (P = 0.1797) | |
| **Age** | | | | | | | | | |
| 50–59 | 10 | 23495.89 | 0.43 | 22 | 96405.22 | 0.23 | 1.87 (0.79–4.10) | 1 | 1 |
| 60–69 | 29 | 20797.03 | 1.39 | 94 | 83199.61 | 1.13 | 1.23 (0.78–1.89) | **3.30 (1.61–6.77)** (P = 0.0011) | **2.462 (1.16–5.222)** (P = 0.0189) |
| 70–79 | 62 | 17398.42 | 3.56 | 188 | 69866.2 | 2.69 | 1.32 (0.98–1.77) | **8.44 (4.33–16.46)** (P < 0.0001) | **7.124 (3.395–14.951)** (P < 0.0001) |
| >80 | 30 | 7051.96 | 4.25 | 125 | 28579.67 | 4.37 | 0.97 (0.63–1.46) | **10.33 (5.05–21.13)** (P < 0.0001) | **7482 (3299–16,971)** (P < 0.0001) |
| **Sex** | | | | | | | | | |
| Male | 23 | 37817.02 | 0.61 | 57 | 152,742 | 0.37 | 1.63 (0.96–2.69) | 1 | 1 |
| Female | 108 | 30926.28 | 3.49 | 372 | 125308.8 | 2.97 | 1.18 (0.94–1.46) | **5.70 (3.64–8.95)** (P < 0.0001) | **3.817 (2.388–6.1)** (P < 0.0001) |
| **Vitamin D prescription** | | | | | | | | | |
| No | 108 | 66601.18 | 1.62 | 380 | 270804.9 | 1.4 | 1.16 (0.92–1.43) | 1 | 1 |
| Yes | 23 | 2142.12 | 10.74 | 49 | 7245.82 | 6.76 | 1.59 (0.92–2.66) | **6.375 (4.06–10.00)** (P < 0.0001) | **3.3 (2.08–5.236)** (P < 0.0001) |
| **Steroid exposure[b]** | | | | | | | | | |
| No steroid | 92 | 41622.03 | 2.21 | 364 | 219780.6 | 1.66 | **1.33 (1.05−1.68)** | 1 | |
| ICS regular use | 1 | 2768.1 | 0.36 | 1 | 688.57 | 1.45 | 0.25 (0.003–19.53) | 0.154 (0.022–1.107) (P = 0.0631) | |
| OCS regular use | 1 | 1097.32 | 0.91 | 1 | 2294.96 | 0.44 | 2.09 (0.03–164.17) | 0.391 (0.055–2.807) (P = 0.3506) | |

Lee et al. (2016), *PeerJ*, DOI 10.7717/peerj.2634

**Table 2** (*continued*)

| Variables | COPD patients | | | Non-COPD patients | | | Incidence Rate Ratio (95 % CI) | Crude HR | Adjusted HR[a] |
|---|---|---|---|---|---|---|---|---|---|
| | Event | PY | Rate | Event | PY | Rate | | | |
| **Smoking-related diagnoses** | | | | | | | | | |
| No | 56 | 26518.7 | 2.11 | 419 | 269633.1 | 1.55 | **1.36 (1.01–1.80)** | 1 | |
| Yes | 75 | 42224.6 | 1.78 | 10 | 8417.64 | 1.19 | 1.50 (0.77–3.24) | 0.848 (0.6–1.20) (*P* = 0.3497) | |
| **Alcohol use disorders** | | | | | | | | | |
| No | 131 | 67001.73 | 1.96 | 429 | 273623.2 | 1.57 | **1.25 (1.02–1.52)** | 1 | |
| Yes | 0 | 1741.57 | NA | 0 | 4427.45 | NA | NA | NA | |
| **Insurance premium (NTD)** | | | | | | | | | |
| <15,000 | 64 | 34007.32 | 1.88 | 257 | 135747.5 | 0.99 | 0.99 (0.74–1.31) | 1 | 1 |
| 15,000—21,999 | 60 | 21653.09 | 2.77 | 158 | 84330.74 | 1.87 | **1.48 (1.08–2.00)** | **3.56 (1.63–7.76)** (***P* = 0.0014**) | 1394 (0.583–3335) (*P* = 0.4554) |
| ≥ 22,000 | 7 | 13082.89 | 0.54 | 14 | 57972.41 | 0.24 | 2.22 (0.76–5.87) | **5.18 (2.37–11.33)** (***P* < 0.0001**) | 1,171 (0.472–2901) (*P* = 0.7338) |
| **Geographic area** | | | | | | | | | |
| North | 54 | 29259.38 | 1.85 | 175 | 121378.3 | 1.44 | 1.28 (0.93–1.75) | 1 | |
| Central | 45 | 18,457 | 2.44 | 141 | 66933.68 | 2.11 | 1.16 (0.81–1.63) | 1.32 (0.89–1.97) (*P* = 0.1658) | |
| South | 29 | 18390.64 | 1.58 | 101 | 79909.28 | 1.26 | 1.25 (0.80–1.91) | 0.856(0.55–1.35) (*P* = 0.50) | |
| East | 3 | 2130.77 | 1.41 | 7 | 6843.38 | 1.02 | 1.38 (0.23–6.03) | 0.76 (0.23–2.43) (*P* = 0.6414) | |
| **Comorbidities** | | | | | | | | | |
| Rheumatoid arthritis | | | | | | | | | |
| No | 129 | 67096.49 | 1.92 | 407 | 272383.3 | 1.49 | **1.29 (1.05–1.57)** | 1 | |
| Yes | 2 | 1646.81 | 1.21 | 22 | 5667.4 | 3.88 | 0.31 (0.04–1.27) | 0.651 (0.161–2.629) (*P* = 0.5463) | |
| Diabetes mellitus | | | | | | | | | |
| No | 102 | 53458.33 | 1.91 | 349 | 216876.3 | 1.61 | 1.19 (0.94–1.48) | 1 | |
| Yes | 29 | 15284.97 | 1.9 | 80 | 61174.37 | 1.31 | 1.45 (0.91-2.24) | 1.015 (0.672–1.533) (*P* = 0.9449) | |
| Hypertension | | | | | | | | | |
| No | 52 | 29753.16 | 1.75 | 149 | 128167.5 | 1.16 | **1.50 (1.07–2.07)** | 1 | |
| Yes | 79 | 38990.13 | 2.03 | 280 | 149883.2 | 1.87 | 1.08 (0.83–1.40) | 1.176 (0.829–1.669) (*P* = 0.3646) | |
| Dyslipidemia | | | | | | | | | |
| No | 81 | 46397.68 | 1.75 | 287 | 188947.6 | 1.52 | 1.15 (0.89–1.48) | 1 | |
| Yes | 50 | 22345.62 | 2.24 | 142 | 89103.05 | 1.59 | **1.40 (1.00-1.95)** | 1.305 (0.917–1.858) (*P* = 0.1386) | |

Lee et al. (2016), *PeerJ*, DOI 10.7717/peerj.2634

**Table 2** (*continued*)

| Variables | COPD patients | | | Non-COPD patients | | | Incidence Rate Ratio (95 % CI) | Crude HR | Adjusted HR[a] |
|---|---|---|---|---|---|---|---|---|---|
| | Event | PY | Rate | Event | PY | Rate | | | |
| Osteoporosis | | | | | | | | | |
| No | 93 | 65071.05 | 1.43 | 297 | 262971.8 | 1.13 | 1.27 (0.99–1.60) | 1 | 1 |
| Yes | 38 | 3672.25 | 10.35 | 132 | 15078.89 | 8.75 | 1.18 (0.80-1.71) | **7.399 (5.071–10.796)** **(P < 0.0001)** | **301 (2.019–4487)** **(P < 0.0001)** |
| Coronary heart disease | | | | | | | | | |
| No | 75 | 50255.5 | 1.49 | 299 | 222287.6 | 1.35 | 1.11 (0.85–1.43) | 1 | 1 |
| Yes | 56 | 18487.79 | 3.03 | 130 | 55763.07 | 2.33 | 1.30 (0.93–1.79) | **2.05 (1.45–2.899)** **(P < 0.0001)** | **1.471 (1.037–2.088)** **(P = 0.0306)** |
| Chronic kidney disease | | | | | | | | | |
| No | 113 | 60552.14 | 2.24 | 375 | 245,400 | 1.83 | 1.22 (0.98–1.51) | 1 | |
| Yes | 18 | 8191.16 | 2.2 | 54 | 32650.65 | 1.65 | 1.33 (0.73-2.30) | 1.222 (0.743–2.010) (P = 0.4303) | |
| Liver disease | | | | | | | | | |
| No | 109 | 56439.49 | 1.93 | 352 | 235095.9 | 1.5 | **1.29 (1.03–1.60)** | 1 | |
| Yes | 22 | 12303.81 | 1.79 | 77 | 42954.81 | 1.79 | 1.0 (0.59–1.62) | 0.944 (0.597–1.493) (P = 0.8051) | |
| Stroke | | | | | | | | | |
| No | 111 | 56270.64 | 1.97 | 343 | 238988.5 | 1.44 | **1.37 (1.10–1.71)** | 1 | |
| Yes | 20 | 12472.65 | 1.6 | 86 | 39062.17 | 2.2 | 0.73 (0.42–1.20) | 0.834 (0.518–1.344) (P = 0.456) | |
| Dementia | | | | | | | | | |
| No | 125 | 65485.98 | 1.91 | 408 | 270683.4 | 1.51 | **1.26 (1.03–1.55)** | 1 | |
| Yes | 6 | 3257.32 | 1.84 | 21 | 7367.3 | 2.85 | 0.65 (0.21–1.66) | 1.003 (0.442–2.277) (P = 0.9934) | |

**Notes.**

Hazard ratio calculation: For fracture sites, comparison was made between COPD and non-COPD; in subcategories such as age and sex, HR is based on the comparison among subcategories with the reference group.

[a]Multivariate Cox regression model derived HR was adjusted for sex, age group, vitamin D prescription, insurance premium category, pre-existing osteoporosis and coronary heart disease.

[b]Only regular users by pre-determined definitions were included for analysis.

Event, Number of Osteoporotic Fractures; PY, Person Years; Rate, Incidence per 1,000 PY; CKD, Chronic kidney disease; HR, Hazard Ratio; ICS, inhaled corticosteroids; IHD, Ischemic heart disease; NA, not applicable; OCS, oral corticosteroids.

Bold type numerals denote statistical significance.

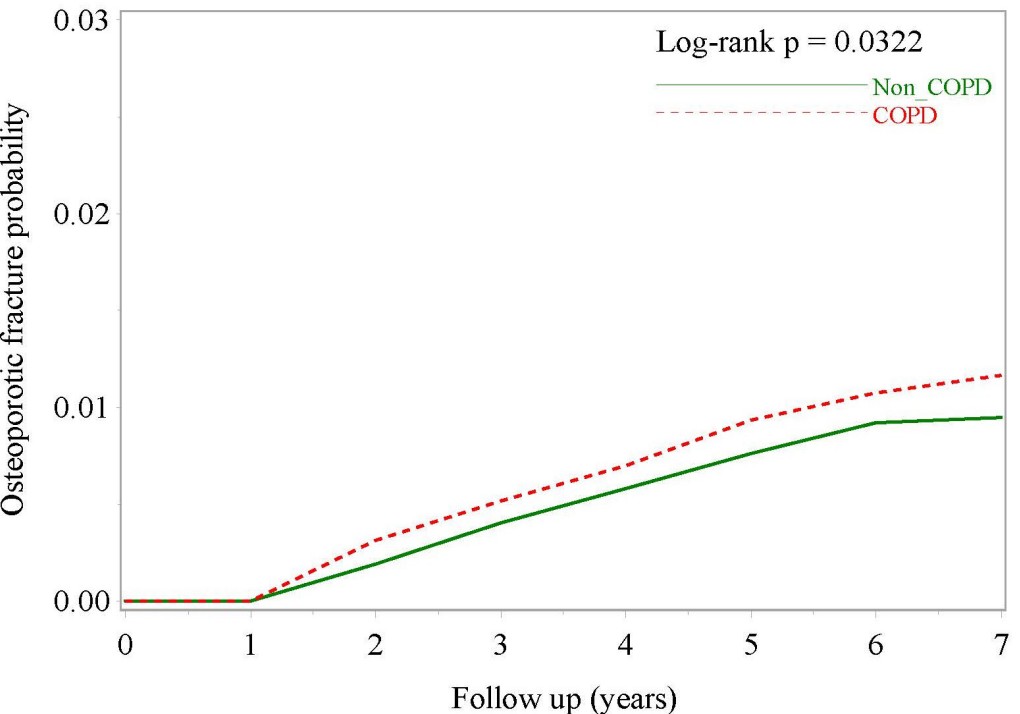

**Figure 2 Cumulative incidence curves of osteoporotic fracture in COPD patients and non-COPD comparators matched for age, sex, index date, osteoporosis, diabetes mellitus and chronic kidney disease.**

dementia, were not associated with OTPF. In addition, steroid exposure, smoking related diagnoses, alcohol use disorders, and geographic region, were not associated with OTPF (Table 2).

## DISCUSSION

We believe this to be the first longitudinal cohort study with a meticulous study design to examine the relationship between COPD and risk of OTPF, particularly vertebral fracture, in an East Asian population. Compared with the comparison cohort matched for age, gender, index month, pre-existing osteoporosis, diabetes mellitus and chronic kidney disease, Asian COPD patients had a significantly increased risk of developing OTPF, with a crude HR of 1.24 (95% CI [1.02–1.51]). When we examine the OTPF by site, the risk for vertebral fractures in COPD patients increases by approximately 30% even after statistical adjustment for confounding factors. In addition to pre-existing osteoporosis, female gender and those who require vitamin D prescription were important clinical predictors of vertebral OTPF in the COPD population.

Chen and colleagues (2015) published a retrospective cohort study to examine the risk for osteoporosis in COPD patients in the same region as ours. Nevertheless, the study reveals some shortcomings: first, pathological fractures resulted from bone metastases,

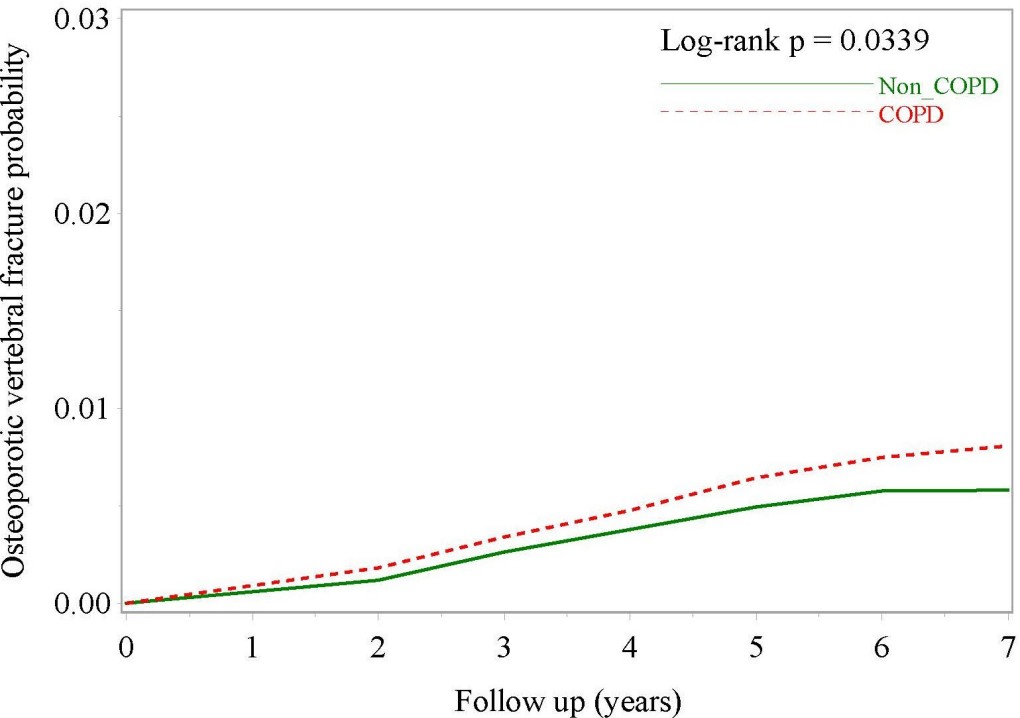

**Figure 3 Cumulative incidence curves of osteoporotic vertebral fracture in COPD patients and non-COPD comparators after multivariate adjustment.**

renal osteodystrophy and hyperparathyroidism were not excluded; second, comparators were not matched for pre-existing osteoporosis, diabetes mellitus, and chronic kidney disease; third, vitamin D prescription was not taken into account; and lastly, a half of the COPD cohort aged below 50 years, which are not an optimal population to examine osteoporosis for. Another population-based retrospective cohort study to examine the association between COPD and hip fracture (*Huang et al., 2016*) revealed that patients with COPD had an approximately 60% greater risk of sustaining hip fracture. Our study cannot confirm this association (crude HR = 2.319; 95% CI [0.68–7.92]; $P = 0.1797$). Further studies are needed to confirm the potentially existing risk.

The prevalence of osteoporosis was significantly higher in COPD patients compared with healthy subjects, with prevalence rates of 9%–69% in COPD patients (*Graat-Verboom et al., 2009*). The association between COPD and osteoporosis has been reported in many studies. In a very elderly Chinese male population, COPD was found to be independently associated with low femoral neck bone mineral density (*Bian et al., 2015*). Higher Global Initiative for Chronic Obstructive Lung Disease (GOLD) stage is reportedly correlated with lower bone mineral density, with every unit decrease in respiratory function (expressed as FEV1 in L/s) associated with a decrease in BMD of approximately 0.02 g/cm$^2$ (*Lekamwasam, Trivedi & Khaw, 2002*; *Lekamwasam, Trivedi & Khaw, 2005*;

Lee et al. (2016), *PeerJ*, DOI 10.7717/peerj.2634

**Table 3   Sensitivity analysis showing the effect of differential time lag on the risk of osteoporotic vertebral fractures among COPD patients who possessed the contributing characteristics.**

| Predictive variable | 2nd follow up year | | | 3rd follow up year | | | 4th follow up year | | | 5th follow up year | | | 6th follow up year | | | 7th follow up year | | |
|---|---|---|---|---|---|---|---|---|---|---|---|---|---|---|---|---|---|---|
| | x/y | A/B | aHR (95% CI) | x/y | A/B | aHR (95% CI) | x/y | A/B | aHR (95% CI) | x/y | A/B | aHR (95% CI) | x/y | A/B | aHR (95% CI) | x/y | A/B | aHR (95% CI) |
| Female | 7,439/ 9,222 | 72/ 17 | **3.3 (1.89– 5.74)**[****] | 6,177/ 7,508 | 48/11 | **3.48 (1.75–6.9)**[***] | 4,909/ 5,956 | 31/6 | **4.46 (1.81– 11.03)**[**] | 3,762/ 4,528 | 20/2 | **8.0 (1.81– 35.4)**[**] | 2,623/ 3,144 | 8/0 | – | 1,444/ 1,726 | 2/0 | |
| Osteoporosis | 959/ 15,702 | 27/62 | **3.31 (2.04– 5.35)**[****] | 758/ 12,927 | 17/42 | **3.35 (1.84–6.09)**[****] | 568/ 10,297 | 10/27 | **3.51 (1.64– 7.51)**[**] | 393/ 7,897 | 7/ 15 | **4.08 (1.61– 10.35)**[**] | 253/ 5,514 | 2/6 | 2.45 (0.48–12.41) | 125/ 3,045 | 1/1 | 595 (0.3–117.73) |
| Vitamin D Prescription | 468/ 16,193 | 17/ 72 | **3.62 (2.1– 6.23)**[****] | 419/ 13,266 | 10/ 49 | **2.97 (1.48–5.97)**[**] | 360/ 10,505 | 4/33 | 1.69 (0.59–4.87) | 295/ 7,995 | 3/ 19 | 1.9 (0.55–6.57) | 212/ 5,555 | 2/6 | 4.48 (0.88–22.9) | 116/ 3,054 | 1/1 | 8.87 (0.46– 170.54) |
| Coronary Heart Disease | 4,590/ 1,2071 | 38/51 | 1.45 (0.95–2.22) | 3,724/ 9,961 | 24/35 | 1.39 (0.82–2.34) | 2,913/ 7,952 | 12/25 | 1.04 (0.52–2.09) | 2,148/ 6,142 | 7/15 | 0.97 (0.39–2.41) | 1,437/ 4,330 | 2/6 | 0.69 (0.14–3.47) | 764/ 2,406 | 0/2 | |

**Notes.**

Bold type numerals denote achieving statistically significant.

[*]$P < 0.05$.

[**]$P < 0.01$.

[***]$P < 0.001$.

[****]$P < 0.0001$.

x/y: number of patients at risk with the characteristic/number of patients at risk without the characteristic.

A/B: number of patients who developed osteoporotic fracture in X/number of patients who developed osteoporotic fracture in Y.

The Cox model was adjusted for sex, age group, vitamin D prescription, and pre-existing osteoporosis.

*Vrieze et al., 2007*). This association was independent of potential confounding factors, such as age, smoking habit, major comorbidity, and medications.

However, only a few studies have examined whether patients with COPD have a higher risk of developing OTPF. A recent small cross-sectional study reported by Watanabe and colleagues reveals that the prevalent vertebral fracture is as high as 79.4% in Japanese men with COPD (*Watanabe et al., 2015*). A case-control study demonstrated an independent association between COPD and increased risk of hip fracture in Catalonians (*Reyes et al., 2014*). In a multicenter cross-sectional study, Diez-Manglano and colleagues observed a high probability of fracture in 347 Spanish COPD inpatients. Overall, nearly half of the COPD patients (95% CI [44.8–54.7]) had a probability of hip fracture in the next 10 years. The probability of fracture was not related to the GOLD stage. An American retrospective observational study of 87,360 COPD veterans reported a high incidence rate of hip fractures during four years of follow-up, with 3.99 events/1,000 person-years (*Morden et al., 2011*). However, this study did not include a comparative cohort for the study population. None of the studies mentioned above used a prospective design method. In a further study with a 6-year follow-up of 5,541 males in the general population, patients with COPD or asthma had lower BMD than patients without COPD or asthma, with adjusted odds ratio for vertebral and non-vertebral fractures of 2.6 and 1.4, respectively (odds ratio for presence of COPD or asthma vs. absence of COPD or asthma, 2.64; 95% CI [1.57–4.44]; and 1.42; 95% CI [1.03–1.96], respectively) (*Dam et al., 2010*). However, the inclusion of patients with asthma limited the ability to evaluate the contribution of COPD to fracture outcomes. Because the abovementioned studies were conducted in the western population and three included men only (*Dam et al., 2010*; *Morden et al., 2011*; *Reyes et al., 2014*), the generalizability to populations in Eastern countries is questionable.

The association between COPD and OTPF observed in the present study may be attributable to shared risks between both conditions, such as smoking, physical inactivity, low body weight, and malnutrition (*Biskobing, 2002*; *Dam et al., 2010*; *Jorgensen & Schwarz, 2008*). Moreover, serum vitamin D levels may also contribute to the risk of OTPF. It has been shown that decreased vitamin D levels were independently associated with increased risk of osteoporosis (defined as vertebral fracture without decreased BMD) by 7.5-fold (*Graat-Verboom et al., 2012*).

It should be noted that OTPF may result from events other than osteoporosis, such as falls. COPD patients typically have many shared risk factors for falls, such as muscle weakness, mobility impairment, and exercise intolerance (*Kim, Mofarrahi & Hussain, 2008*; *Ries et al., 2007*). Compared with healthy controls, the presence of COPD has been shown to be associated with significantly impaired ability to perform balance tests, such as the Berg Balance Scale, timed up and go test, and single-leg stance (*Crisan et al., 2015*; *Porto et al., 2015*). An observational cohort study reported a high prevalence of previous falls in COPD patients (32%), with an incidence rate of 0.1 (95% CI [0.06–0.14]) falls/person-month (*Roig et al., 2011*). In a population-based study, Sibley et al. reported that COPD was significantly associated with increased risk of falls (*Sibley et al., 2014*). Thus, in addition to osteoporosis, risk factors for falls may play an important

role in the correlation between COPD and OTPF. The treatment of such risk factors and osteoporosis is likely required for successful fracture prevention.

Patients with CKD are at an increased risk of fragility fracture (*Nickolas, Leonard & Shane, 2008*), which may be explained by CKD-related bone mineral disease and high co-prevalence of CKD and osteoporosis in elderly individuals (*Klawansky et al., 2003*). This is the reason why our study design employed matching the comparison group by this factor.

The present study had some clinical implications. According to current guidelines, osteoporosis screening is not recommended for patients with COPD. In the present study, we identified COPD as a potential risk factor for OTPF, particularly vertebral OTPF and the contributing predictors of vertebral OTPF are female COPD patients or those with pre-existing osteoporosis or required vitamin D prescription. Timely arrangement of BMD scanning may increase the possibility of treating osteoporosis and reduce the risk of fracture in this high-risk population. Systemic corticosteroid should be used with cautious as the incidence of fracture increases with duration and dosage of steroid therapy (*Canalis et al., 2007*; *Reid & Heap, 1990*). Because 87%–90% of elderly fractures result from falls (*Dargent-Molina et al., 1996*; *Fife & Barancik, 1985*), strategies to prevent falls should be implemented in at-risk patients, such as medication review, environmental adjustment, and exercise training. Previous studies have shown that balance training in COPD patients has utility in improving important fall-related factors, such as function, muscle strength, and balance performance (*Beauchamp et al., 2013*; *Harrison et al., 2015*). Thus, introducing balance training component into currently existing exercise programs for COPD patients may be of more value in OTPF prevention.

Several limitations of the present study have to be acknowledged. First, we were unable to obtain data regarding important fracture risk factors, including BMI, bone marrow density, and calcium intake, family history of OTPF, and physical activity level. Second, the propensity for falls could not be determined because data for risk factors, such as gait abnormality, visual impairment, and living environment, were unavailable. Third, COPD diagnoses were obtained from a claimed dataset, which might not be as accurate as diagnoses made by standardized protocols or tools. However, we excluded outpatients claiming less than two times per year without an inpatient record to increase accuracy. Lastly, the true incidence of prevalent fracture which is existent morphological fracture identified by x-ray during the study period cannot be determined. In the current study, the authors only picked up ''clinical fractures'' with symptoms because most patients were not taking screening spinal x-ray exams. Thus, upon subject recruitment, a prevalent asymptomatic vertebral fracture can potentially be missed. Similarly, the fracture that this research was looking at was physician-diagnosed clinical fracture identifiable in the national claims database. Thus, the true vertebral fracture incidence during the study period is unknown. Thanks to the nationwide frailty intervention health programs established in Taiwan aiming to reduce the risk of falls and fractures, many if not all asymptomatic pre-clinical vertebral OTPFs have already been screen-detected. The risk of misclassification bias is thus much reduced.

The study has several strengths. First, we used compulsory administrative data to identify our study cohorts, which avoids possible volunteer or selection bias. Second,

patients in the COPD group of the present study were meticulously matched with individuals without COPD, thereby minimizing possible confounding or bias that may affect other observational studies. Third, the sample of the present study was large and nationwide, increasing the generalizability of the present findings. We believe the effect of the large sample size of the present study may overcome some of the confounding factors caused by minor issues, such as code misclassification and minor differences between the two cohorts or use of medications that effect bone metabolism. Fourth, we excluded patients with fractures that occurred within the first year of COPD diagnosis, which may have been caused by other factors as the time was too short for COPD to have affected bone metabolism.

In conclusion, the present large-scale longitudinal study conducted in the Asia-pacific region found an association between OTPF in COPD patients. Among patients with COPD, the risk of fractures was significantly increased in female patients, and those with a diagnosis of osteoporosis, or those who require vitamin D prescription indicating risk for osteoporosis. These results indicate the importance of fall prevention and osteoporosis treatment in patients with COPD. Further prospective cohort studies are required to confirm the causal relationship between COPD and OTPF.

## ACKNOWLEDGEMENTS

The authors wish to thank the Administration of the National Health Insurance and the National Health Research Institute, Taiwan, for the dataset used to conduct this research. The interpretation and conclusions contained herein do not represent those of the institutions above.

### Funding

The authors received no funding for this work.

### Competing Interests

The authors declare there are no competing interests.

### Author Contributions

- Ping-Hsueh Lee analyzed the data, wrote the paper, reviewed drafts of the paper.
- Victor C. Kok conceived and designed the experiments, performed the experiments, analyzed the data, wrote the paper, prepared figures and/or tables, reviewed drafts of the paper, applied for Ethics review.
- Po-Liang Chou performed the experiments, analyzed the data, prepared figures and/or tables, reviewed drafts of the paper.
- Ming-Chang Ku and Yu-Ching Chen analyzed the data, prepared figures and/or tables, reviewed drafts of the paper.
- Jorng-Tzong Horng performed the experiments, analyzed the data, contributed reagents/materials/analysis tools, reviewed drafts of the paper.

## Human Ethics

The following information was supplied relating to ethical approvals (i.e., approving body and any reference numbers):

Kuang Tien General Hospital Institutional Review Board.

approved this study: KTGH-IRB 10520.

## Data Availability

Chou B. 2016. [raw data]revised_match_case_control.sas7bdat. Figshare: DOI 10.6084/m9.figshare.3826656.v1.

## Supplemental Information

Supplemental information for this article can be found online at http://dx.doi.org/10.7717/peerj.2634#supplemental-information.

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
