# Peer review of "Risk and clinical predictors of osteoporotic fracture in East Asian patients with chronic obstructive pulmonary disease: a population-based cohort study"

_PeerJ, doi:10.7717/peerj.2634_

## Round 0.1 · original submission · Major Revisions

This is a potentially interesting manuscript. Some methodological concerns are here raised, and significantly reduced the editor’s enthusiasm in the manuscript. Therefore, in its current form, the manuscript is not considerable acceptable for publication.

Major points:

1. This observational, population study in my humble opinion is not a prospective or retrospective cohort study, because the subjects in the two groups were not comparable (table 1) before the exposure was given.

2. Some potential confounders may be present, but not included in the analyses, including vit D levels, bone mineral density, and pro-osteoporosis and anti-osteoporosis medications. [1-5] It is interesting that another study using the same National Health Insurance Research Database could extract data of steroid use and other factors.[6]

3. The control group should have been the patients with matched co-morbidities such as diabetes, CKD and CHD (see also point #1). Table 1 shows, clearly, that the control and case groups had different frequencies of these co-morbidities (lower in the control group). The comparison therefore is less sound. The follow-up years were also different between the two groups, which may be attributable to the deaths and lost to follow-up and contributed to the difference.

4. Multivariate analyses should only include the factors that were revealed significant in univariate analyses (tables 2 and 3). Otherwise, many noises may have been brought in.

5. Sensitivity (subgroup) analyses are also recommended to test various co-variates, particularly the co-morbidities and other known factors associated with fracture and/or osteoporosis.

6. It is recommended to subcategorize the factures into vertebral and non-vertebral, or other more detailed groups.

7. Some important reports should have been cited, particularly given the much similarity between this study and some published works. [1, 2, 6-10] More importantly, the authors must justify what is new in the study or how this study fills in the knowledge gap in the field, although the PeerJ does not reject articles based on novelty.

8. It is interesting to probe the osteoporotic fracture free survival; however, it would be more meaningful in my view to plot the probability of osteoporotic facture and time (event would be osteoporotic facture, versus death or osteoporotic fracture in current model).

Minor points:

1. In the abstract, the expression of “aHR was non-significantly increased” seems not in line with standard English and needs a modification. A careful review of the manuscript

is also recommended to address the concerns over its writing style. Language service or review by a native English speaker is also recommended.

2. What is the outcome of the HR calculated for tables 2 and 3? In my humble view, HR is only computed for survival analysis. The author probably meant odds ratio?

3. Please provide STROBE check list since this is an observational study.

References

[1] Inoue D, Watanabe R, Okazaki R. COPD and osteoporosis: links, risks, and treatment challenges. International journal of chronic obstructive pulmonary disease. 2016;11:637-48.

[2] Graat-Verboom L, Smeenk FW, van den Borne BE, Spruit MA, Donkers-van Rossum AB, Aarts RP, et al. Risk factors for osteoporosis in Caucasian patients with moderate chronic obstructive pulmonary disease: a case control study. Bone. 2012;50:1234-9.

[3] Franco CB, Paz-Filho G, Gomes PE, Nascimento VB, Kulak CA, Boguszewski CL, et al. Chronic obstructive pulmonary disease is associated with osteoporosis and low levels of vitamin D. Osteoporosis international : a journal established as result of cooperation between the European Foundation for Osteoporosis and the National Osteoporosis Foundation of the USA. 2009;20:1881-7.

[4] Biskobing DM. COPD and osteoporosis. Chest. 2002;121:609-20.

[5] Goldstein MF, Fallon JJ, Jr., Harning R. Chronic glucocorticoid therapy-induced osteoporosis in patients with obstructive lung disease. Chest. 1999;116:1733-49.

[6] Huang SW, Wang WT, Chou LC, Chen HC, Liou TH, Lin HW. Chronic Obstructive Pulmonary Disease Increases the Risk of Hip Fracture: A Nationwide Population-Based Cohort Study. Scientific reports. 2016;6:23360.

[7] Liao KM. Chronic obstructive pulmonary disease is a strong independent risk factor for osteoporosis and pathologic fractures: a population-based cohort study. QJM : monthly journal of the Association of Physicians. 2016;109:69.

[8] Kwok TC, Su Y, Khoo CC, Leung J, Kwok A, Orwoll E, et al. Predictors of non-vertebral fracture in older Chinese males and females: Mr. OS and Ms. OS (Hong Kong). Journal of bone and mineral metabolism. 2016.

[9] de Miguel-Diez J, Jimenez-Garcia R, Hernande-Barrera V, Carrasco-Garrido P, Puente-Maestu L, Ramirez Garcia L, et al. Is COPD a Risk Factor for Hip Fracture? Copd. 2016:1-11.

[10] Chen SJ, Liao WC, Huang KH, Lin CL, Tsai WC, Kung PT, et al. Chronic obstructive pulmonary disease and allied conditions is a strong independent risk factor for osteoporosis and pathologic fractures: a population-based cohort study. QJM : monthly journal of the Association of Physicians. 2015;108:633-40.

Reviewer 1 ·

Basic reporting

No comments.

Experimental design

According to the authors, subjects with pre-existing COPD or OTPF before January 1, 2007 were excluded. However, because what matters here seems to be the time to first fracture after the index date (the time point when COPD was newly diagnosed), the authors should exclude subjects with OTPF before the index date.

Validity of the findings

1) As shown in Table 1, one major problem is that nearly a half of OTPF occurred in patients who had pre-existing osteoporosis, although they were about only 10% of the population either in COPD or non-COPD group. First, what was the impact of COPD in each population (those with pre-existing osteoporosis and those without) on fracture? Second, information about anti-osteoporotic medications should be included in the analysis.
2) I wonder if COPD is associated with increased fracture rate if the analysis were restricted to males with or without multiple adjustments.

Additional comments

1) This is a retrospective cohort study and should be explicitly described so in the abstract.
2) In the abstract, the phrase "followed up for 7 years" seems misleading. The mean follow-up period sounds more appropriate here.
3) In Table 2, rate ratio should be shown with 95% CI.
4) Fracture sites in each group should be shown. The authors could at least show some information about #ICD code.
5) The authors describe (line 105) that to date no studies have been reported from East Asian countries regarding the association between COPD and OTPF. While it seems true that there have been very few longitudinal studies investigating clinical fracture, there are some cross-sectional studies including, for example, a Japanese study by Watanabe R et al, which analyzed COPD male patients for prevalent fracture. The authors should cite such papers and discuss the issue of clinical versus prevalent (vertebral) fracture. "Fracture" in the current analysis should be mostly, but not all, "clinical fracture." The fact that discrimination between clinical and prevalent fracture may even be difficult is another limitation.
6) Discussion is somewhat lengthy and redundant. Etiology and mechanisms are particularly irrelevant here, because this study did not analyze risk factors and thus provides virtually no insight into the mechanism of COPD-associated osteoporosis.

---

## Round 0.2 · Minor Revisions

Please consider and respond to the constructive reviewer comments.

Reviewer 1 ·

Basic reporting

good

Experimental design

good

Validity of the findings

good

Additional comments

The revision appears to have been mostly well done except for a couple of points as follows:
1) As for the authors' response to the previous comment 5) "The fact that discrimination between clinical and prevalent fracture may even be difficult is another limitation,"
there seems some misunderstanding here. Prevalent fracture is existent morphological fracture identified by x-ray. In the current study the authors basically only picked up "clinical fractures" with symptoms because most patients were not taking regular spinal x-ray exams. Thus, upon subject recruitment the authors could exclude subjects with the past history of clinical fracture but presumably missed most subjects with prevalent asymptomatic vertebral fracture. Similarly, the fracture that the authors were looking at in this study was clinical fracture identifiable in the national claims database. So the true vertebral fracture incidence during the study period is unknown. When discussing vertebral fracture, it seems important to clearly discriminate morphological and clinical fractures (i.e., x-ray-based or not).
2) In the revised table 2, in COPD patients 131 evens occurred in 68,743 PY, but there are only 94 altogether in "steroid exposure" subcategories. And "smoking" and "alcohol" subcategories include 202. The numbers of events in non-COPD subjects and PY also seem different from the other subcategories. Please carefully re-review all the numbers in the table and make corrections if necessary.
3) In Table 2, for fracture sites comparison was made between COPD and non-COPD. But in the other subcategories, HR is based on the comparison among subcategories with the reference group and restricted to COPD subjects (if I am right). This is not very easily understandable and may even be somewhat misleading. More explanations would be necessary in the legends.

---

## Round 0.3 · accepted · Accept

Thank you for submitting your work to PeerJ!

Reviewer 1 ·

Basic reporting

good

Experimental design

good

Validity of the findings

good

Additional comments

good